# Differential effects of habitat loss on occupancy patterns of the eastern green lizard *Lacerta viridis* at the core and periphery of its distribution range

**Ana María Prieto-Ramirez**[1]*, **Leonie Röhler**[2], **Anna F. Cord**[2], **Guy Pe'er**[3,4,5], **Dennis Rödder**[6], **Klaus Henle**[1]

**1** Department of Conservation Biology, Helmholtz Center for Environmental Research–UFZ, Leipzig, Germany, **2** Department of Landscape Ecology, Helmholtz Center for Environmental Research–UFZ, Leipzig, Germany, **3** German Center for Integrative Biodiversity Research (iDiv) Jena-Halle-Leipzig, Leipzig, Germany, **4** Department of Economics and Department Ecosystem Services, Helmholtz Center for Environmental Research–UFZ, Leipzig, Germany, **5** University of Leipzig, Leipzig, Germany, **6** Department of Herpetology, Zoological Research Museum Alexander Koenig ZFMK, Bonn, Germany

* ana-maria.prieto-ramire@ufz.de

**Data Availability Statement:** Data are available at: https://github.com/anaprieto1/Occupancy-of-Lacerta-viridis-.

## Abstract

The effects of habitat loss on the distribution of populations are often linked with species specialization degree. Specialist species can be more affected by changes in landscape structure and local patch characteristics compared to generalist species. Moreover, the spatial scale at which different land covers (eg. habitat, cropland, urban areas) affect specialist species can be smaller. Specialization is usually assumed as a constant trait along the distribution range of species. However, for several taxa, there is evidence of higher specialization degree in peripheral populations compared with populations in the core. Hence, peripheral populations should have a higher sensitivity to habitat loss, and strongest effects should be found at a smaller spatial scale. To test these expectations, we implemented a patch-landscape approach at different spatial scales, and compared effects of landscape structure and patch characteristics on occupancy probability among northern peripheral, more specialized populations (Czech Republic) and core populations (Bulgaria) of the eastern green lizard *Lacerta viridis*. We found that landscape structure and patch characteristics affect differently the occupancy probability of *Lacerta viridis* in each region. Strongest effects of habitat loss were found at a spatial scale of 150m around patches in the periphery, but at a scale of 500m in the core. In the periphery occupancy probability of populations was principally affected by landscape composition, and the effect of habitat quality was stronger compared to core populations. In the core, persistence of populations was mainly explained by characteristics of the spatial configuration of habitat patches. We discuss possible ecological mechanisms behind the relationship between sensitivity to habitat loss, populations' specialization degree and position in the distribution range, and suggest conservation measures for *L. viridis*.

**Funding:** A.M.P.R. German Society for Herpetology and Herpetoculture DGHT Hans-Schiemen-Fonds https://www.dght.de/startseite Heinrich Böll Foundation HBS Scholarship number P113742 https://www.boell.de/de/stiftung/heinrich-boell The funders had no role in study design, data collection and analysis, decision to publish, or preparation of the manuscript.

**Competing interests:** The authors have declared that no competing interests exist.

## Introduction

Anthropogenic land-use changes lead to the loss of natural and semi-natural habitats, resulting in reduced overall amount of habitat available, fragmentation into smaller patches and increasing isolation among these patches due to land-use intensification forming a matrix of inhospitable land. These processes alter landscape composition and configuration: as patch area decreases, patch isolation increases, and spatial relations between landscape elements (e.g. habitat, non-habitat areas, and topographic features like rivers) are altered. The ecological consequences for species, at the landscape scale, include reduced functional connectivity and reduced viability [1], leading to declining trends in abundance and distribution.

The effects of modified landscape structure on the distribution of natural populations have been widely studied and linked with species-specific traits [2–4]. In particular, habitat specialization is one of the main traits shaping species' response to habitat loss [5, 6]. Specialist species are known to be more sensitive to changes in patch size [7–9], isolation [10–12], habitat quality [13], and overall amount of habitat in the landscape [14, 15], whereas generalist species can typically better cope with reduced patch size and overall reduce in the amount of habitat [16].

Differential responses to habitat loss between generalist and specialist species have also been linked to the 'scale of effect' of different parameters. We define the 'scale of effect' as the extent of area at which the strongest effect of a given factor on an ecological response is found [17]. It has become a central topic in ecology in the past years, with particular focus onto the question how landscape composition influences species' distribution. The scale of effect of habitat amount on species' distribution has been shown to be smaller for specialist than for generalist species across different taxa such as butterflies [18], birds [14, 16] and rodents [19]. Similarly, the scale of effect of other landscape composition variables is usually expected to be smaller for specialist species [20].

Studies on the effects of habitat loss that consider species' specialization usually assume species to be characterized by the same trait along their distribution range. However, the degree of specialization can change across the distribution range of a given species, resulting in intraspecific differences among populations. The Kühnelt principle [21] states that the range of colonizable habitats is wider at the core of the distribution range where environmental conditions are optimal, whereas at the periphery conditions are suboptimal and fewer microhabitats are suitable for the species. Therefore, populations at the core should be habitat generalists ("euryoecious"), while populations at the periphery of the species' range can be, in comparison, more specialized ("stenoecious") [22]. Accordingly, it has been found in lizards [23, 24], birds [25] and insects [26] that individuals in peripheral populations have narrower realized niches than those living in the core of the distribution range. However, in spite of existing evidence, most studies on habitat loss carried out at broad scales, involving the total or partial extent of the distribution range of a species, have overlooked this variability, and therefore, the possible differential effects on distribution patterns. Consequently, conservation measures applied at local scales–especially in the periphery–might not be adequate enough to protect threatened populations if the measures were derived from analyses of habitat loss effects in other parts of the distribution range.

Here we investigated the effects of habitat loss and fragmentation on the occupancy patterns of core and northern peripheral populations of the eastern green lizard *Lacerta viridis*. Recently, it has been found that northern peripheral populations of *L. viridis* (Germany, Czech Republic) have a higher specialization degree compared to core populations (Bulgaria) [24]. In the periphery, populations have narrower niches and can only persist in habitats with comparably lower vegetation structure that allow them to compensate for suboptimal overall climatic conditions (e.g. lower radiation). In the core, populations have a broader range of available

habitats and select for microhabitats with higher vegetation structure. The higher specialization degree of *L. viridis* populations in the northern periphery suggests that these populations might also have a higher sensitivity to habitat loss and fragmentation compared to generalist populations living in the core of the distribution range.

In this study, we implemented a patch-landscape approach to evaluate the occupancy patterns of populations of *L. viridis* in Bulgaria (core) and in the Czech Republic (periphery). Our main objectives were to find out which are the most relevant spatial scales affecting patch occupancy in each region and which parameters of the landscape structure and patch characteristics have the strongest effect. We expected to find at the intraspecific level the same patterns of the effects of habitat loss reported at the species level. We hypothesized that: 1) the relevant scale(s) at which occupancy is best explained should be smaller at the periphery compared to the core; 2) the proportion of different land-cover types will have a smaller scale of effect at the periphery compared to the core; and 3) peripheral populations are more sensitive to isolation, area and reduced habitat quality compared to core generalist population.

## Methods

### Study areas

The study regions were located in the northern periphery and in the core of the distribution range of *L. viridis* (Fig 1). The study region at the species' periphery was located in the surroundings of Prague (Bohemia, Czech Republic), where populations are located in open stony areas within open oak forest and along the cliffs of the Moldova valley, as well as those of other valleys perpendicular to the Moldova river valley (Pr; Fig 1B). The core region was located in the Thracian Plain of Bulgaria, in the surroundings of Plovdiv (Core; Fig 1C). The region is an alluvial plain dominated by the banks of the Maritsa River and its tributary rivers. Here *L. viridis* inhabits diverse natural and semi-natural habitats, from road edges and open shrubland to mesophilic forest [27]. In both study regions habitat of *L. viridis* has been lost due to agricultural expansion and intensification, as well as by (semi-)urban development. We selected landscapes in both regions with similar configuration and composition characteristics that could ensure enough levels of comparability. Both landscapes had low percentages of habitat (11.2% in the core and 13.1% in the periphery) and similar habitat configuration in terms of ranges of patch area and isolation (S1 Appendix).

### Field survey

Field surveys were carried out in Plovdiv in 2014 and in Prague in 2015. *L. viridis* is active from beginning of April to beginning of October in Bulgaria, and from late May to beginning of September in the Czech Republic. Therefore, in order to make surveys comparable, data collection was carried out earlier in the core than in the periphery: From beginning of April to late May in the core, and from mid-May to late July in the periphery. The difference in sampling times made average maximum air temperatures per sampling month relatively similar among regions (Core: 18.5–23.4˚C; periphery: 22.5–– 24.6˚C).

Based on literature about the habitat requirements of *L. viridis*, and available information about places where the species has been found in each region (pers.com: Plovdiv: Tzankov, N; Prague: Moravic, J; Chamlar, J.), we identified patches of habitat to be surveyed in each region using satellite maps available in Google earth. We visited 42 patches in the core and 33 in the periphery (see S2 Appendix for locations). All polygons corresponding to the edges of the surveyed patches in both regions were manually digitalized using ArcMap [29].

Occupancy surveys and analysis were designed following the protocol proposed by Mackenzie and Royle [30], prescribing a specific number of visits depending on the probability of

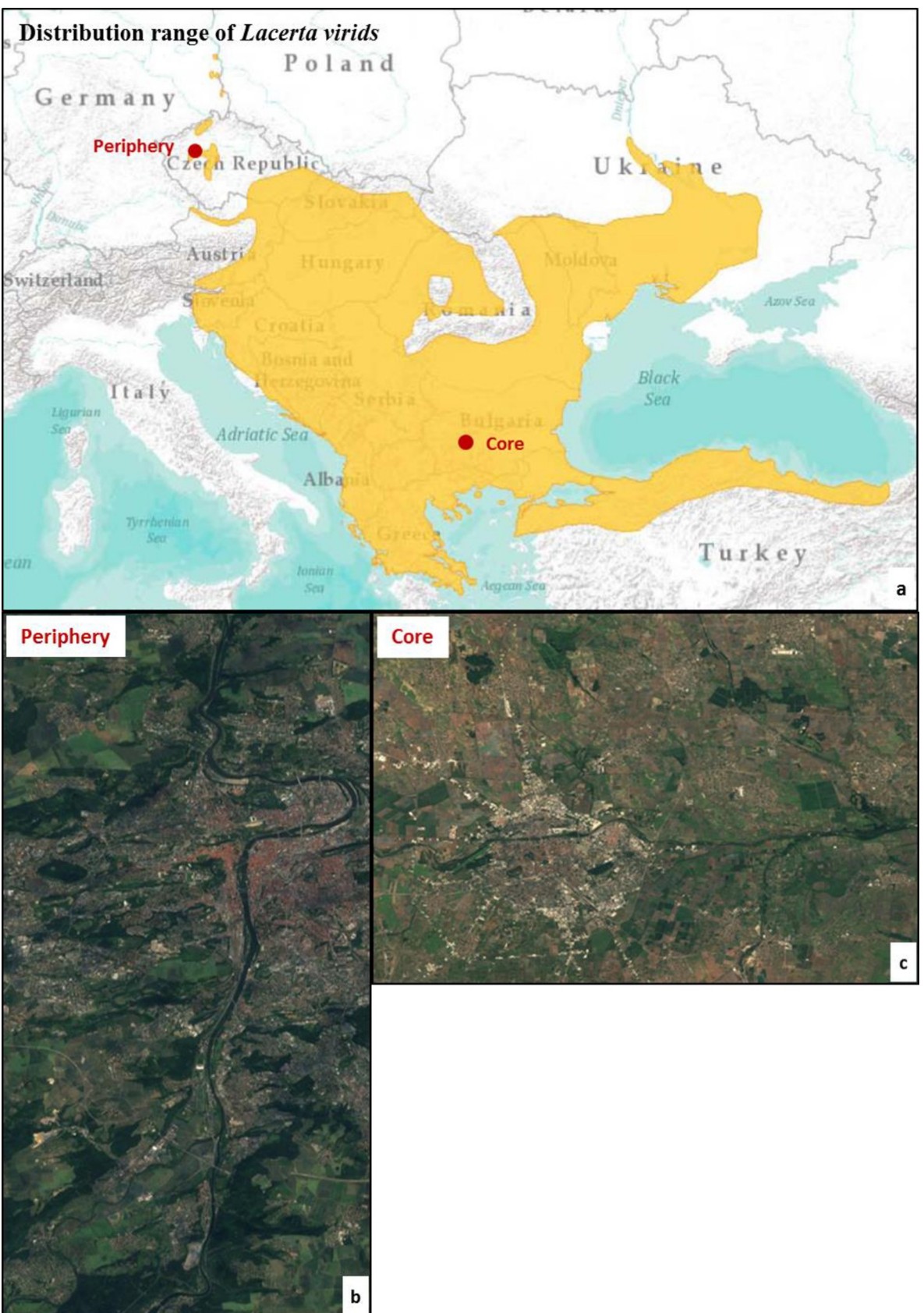

**Fig 1.** Distribution range of *Lacerta viridis* (a) and location of the study sites. In the periphery (b) the study site is located in the surroundings of Prague and has an extent of 522 km$^2$ (location: top-left 50.17˚N, 14.29; top-right 50.16˚N, 14.46˚E; bottom-left 49.92˚N, 14.27˚E; bottom-right 49.92˚N, 14.45˚E). The study site in the core region (c) corresponds to the surroundings of Plovdiv and has an extent of 325 km$^2$ (location: top-left 42.26˚N, 24.68˚E; top-right 42.24˚N, 24.93˚E; bottom-left 42.12˚N, 24.66˚E; bottom-right 42.10˚N, 24.91˚E). Images source: a: IUCN, *Lacerta viridis* distribution range [28]; b and c: Sentinel-2 cloudless 2016 by EOX IT Services GmbH CCBY 4.0 license.

detection of the species. Based on estimates of detection probability for similar species [31, 32], the number of surveys per patch was set to two, one in the morning (9:00–12:00 a.m.) and one in the afternoon (14:00–19:00 p.m.) of the same day or one day later, in accordance with the species' daily activity pattern [33].

Surveys lasted one hour each, walking along predetermined line transects. With a standard walking speed of 20 m/min, which is slow enough to search and detect lizards, a one hour survey corresponds to a total length of 1200m, which were subsequently divided into transects. As most patches had a heterogenous compostion, the number and length of transects varied depending on the number of different habitat types into each patch and the proportion of area of the patch covered by each habitat type. Nevertheless, all the transects in a patch always summed up 1200 m to assure one hour visit. Satellite imagery was used to define the relative coverage of each habitat type within each patch. Transect lengths varied between 50–400 m. Transects were located at least 100 m of each other, and the total length of each transect was placed in only one habitat type. The number of transects surveyed per patch ranged from three to 12. During transect walking, a width of 2.5 m was scanned at each side of the transect to visually search for *L. viridis*. As surveys were based on visual identification of lizards, and no collection of biological material or handling of animals was required, no permits were necessary for carrying out this study.

## Land cover classification

To calculate landscape composition variables around each patch (see section "Calculating patch variables and landscape structures" below), we generated land-cover maps for the two study sites. Land cover classes in each region are described in Table 1. Based on reported literature, we define habitat types as the different vegetation structures used by *L. viridis* in each region. Relevant habitat types in the core were: woodland, shrubland, rocky outcrop vegetation (rocky_veg), grassland, transitional vegetation (trans_veg) and open ground and river beds (bare soil). Habitat types in the periphery were: open woodland (openwood), shrubland, rocky outcrop vegetation, dry grassland (dry_grass) and transitional vegetation. Natural or semi-natural areas that are non-habitat in the periphery were dense woodland (densewood) and humid grassland (humid_grass). In both regions, urban areas (urban), and crops and pastures (crop_pas) were defined as other non-habitat land-cover classes (S3 Appendix).

To obtain the land cover classified map in the core, a supervised Mahalanobis Distance classification of cloud free, atmospherically and topographically corrected Rapid Eye satellite imagery (acquired on May 8[th], 2014; 5m resolution), in combination with information derived from the Copernicus Land Monitoring Service (incl. Urban Atlas 2012, Imperviousness Degree–IMD 2012 and Tree Cover Density–TCD 2012; 20m resolution) was performed. Training (polygon) data for the target classes were generated based on land cover information collected during the field survey and complemented by data digitized based on the RapidEye imagery. Post-processing included a majority analysis (except for the class urban) with a kernel size of 3x3 to remove isolated cropland pixels mapped within (semi-)natural vegetation cover. The final map had an overall accuracy of 91.1%. All processing and analyses were performed in ArcGIS 10.6 [34] and ENVI 5.0 [35]

**Table 1. Land cover classes conforming the classified maps of both, core and periphery, regions.**

| Land cover class | Variable name | Description | Region | Habitat |
|---|---|---|---|---|
| Bare soil | Bare_soil | Open ground corresponding to not paved ways in the interior of patches and sandy, not vegetated river beds | Core, Periphery | Yes |
| Rocky outcrop vegetation | Rocky_veg | Rock outcrops and its associated grasses and herbs | Periphery | Yes |
| Grassland | Grass | Dry and mesic grasslands | Core | Yes |
| Dry_grassland | Dry_grass | Broad leaved dry grassland, termophilus herbs, ecotones at the edge of forest and shrubs | Periphery | Yes |
| Humid grassland | Humid_grass | Perennial grasses in wetlands, wet meadows, moor grasses and river bed grasslands and herbs | Periphery | No |
| Shrubland | Shrubland | Shrubs and scrubs areas | Core, Periphery | Yes |
| Transitional vegetation | Trans_veg | Transitional woodlands with cover density <30% | Core, Periphery | Yes |
| Woodland | Woodland | Woodland with crown cover density >30% | Core | Yes |
| Open woodland | Openwood | Woodland with crown cover density between 30%– 75% | Periphery | Yes |
| Dense woodland | Densewood | Woodland with crown cover density between 75% and 100% | Periphery | No |
| Crops and Pastures | Crop_pas | Areas used for agricultural activities, either cultivation or pasture purposes | Core, Periphery | No |
| Urban area | Urban | Continuous and discontinuous urban fabric, road networks | Core, Periphery | No |

Classification of land cover classes in the periphery was achieved by reclassifying the most recent vegetation community and land-use map [36] available from the Prague Institute for Planning and Development (Institut plánování a rozvoje hl. m. Prahy, IPR). This is a vector map with 5m resolution with 66 classes: 10 corresponding to different urban land uses, two to agriculture and pastures, and 52 representing different vegetation communities. In a first step we reclassified the vegetation communities that correspond to dry_grass, humid_grass, shrubland, rocky_veg and woodland. In a second step, woodland was reclassified as openwood, densewood and trans_veg based on tree cover density (TCD) data available from CORINE. Areas in the northern and southern edges of that study site were unfortunately not covered by the IPR maps. Therefore, for these areas we produced a land cover map based on the Urban Atlas 2012 and TCD information, and when necessary, manually digitalized the different classes by using orthophotos available from the IPR webpage.

## Calculating patch variables and landscape structures

To evaluate the possible differential effects of habitat loss in the core and periphery, we applied a patch-landscape approach and analyzed the influence of variables representative of landscape structure and patch characteristics on occupancy. We differentiated between four types of variables: landscape configuration, landscape composition, patch geometry and patch habitat quality. Variables defining the landscape configuration around each patch included distance to river (dist_river), distance to urban areas (dist_urban) and distance to crops and pastures (dist_crop), and two measures of isolation, the edge-to-edge Euclidean distance to the nearest patch (np_dist) and proximity index (prox).The proximity index (Gustafson and Parker, 1994) is a scale dependent measure of isolation and is calculated as the sum of the ratios patch area / distance to the focal patch for all patches that fall, at least partially, into the buffer of a given distance around the focal patch.

Variables related to landscape composition were calculated at different buffer-distances (hereafter, "scales") around each patch in each region. The different scales were selected based on reported dispersal distances for *L. viridis* [37–39]. Scales selected were: 50m, 150m, 250m,

500m, 750m, 1km, 1.5km, 2km, 2.5km and 3km. At each scale, we calculated the proportion of urban, crop_pas and habitat (the sum of all habitat types).

Patch geometry variables included area, perimeter, perimeter to area ratio (Per_area) and shape index (Shape_index). Patch habitat quality was defined based on the most important parameters found for this species [24, 40–42]: vegetation structure, radiation and slope. Vegetation structure was calculated based on available information at the microhabitat scale. At each single transect in each patch, percentage of vegetation coverage was taken in at least one plot of 25 m$^2$. Vegetation coverage classes included herbs < 30 cm, herbs between 40 and 80 cm, herbs > 90 cm, woody plants < 2 m, woody plants > 2 m, dry leaves, rocks and fallen trunks, bare soil, and branches coverage. Plots correspond either to the area around the specific point where a lizard was detected or to the area around random points blindly selected in the GPS along each transect. For each plot we calculated the foliage height diversity' index (FHD; [43]), which is a modification of the Shannon index applied to vegetation structure. Because most of the patches had a heterogeneous habitat composition, the plots of a single patch might belong to different habitat types. Therefore, we averaged the FHD values of the plots belonging to the same habitat type across patches to obtain the averaged FHD of each habitat type. Vegetation structure (Veg_str) of each patch was then calculated as the sum of the FDH of each habitat type weighted by the area that each specific habitat type occupied within the patch. To calculate the topographic slope we used software SAGA [43] to derive slope maps from digital elevation models (DEMs) with 30m resolution available from the U.S Geological Survey. We averaged pixel values corresponding to each patch. We calculated radiation from the DEMs with the 'Potential incoming solar radiation' module of SAGA [44]. Radiation value of each patch hence corresponded to the average annual radiation during the 5 years preceding the field work in each region, calculated from April to September, from 8am to 6pm and with a temporal resolution of 10 days and two hours. All other calculation procedures were carried out with ArcMap version 10.3.1 [28], except for shape_index and prox which were calculated with FRAGSTATS version 4 [45].

## Statistical analysis

To evaluate the occupancy patterns of populations of *L. viridis*, we applied the occupancy model proposed by MacKenzie and Bailey [46] as implemented in the package 'Unmarked' [47] in the software R [48]. This model calculates the probability of occupancy (*p*) by correcting for the probability that an individual will actually be detected (*psi*). The first step was to fit a detection probability model to be used in all subsequent steps. For this, we tested the effect of vegetation structure, day of survey and patch area on detection probability. As previously shown, vegetation structure can affect the detectability by reducing the visibility for the observer. Day influences lizards' activity, given it is determined by annual seasonality, increasing with the advance of the spring and starting to decrease at the beginning of the summer in the core, and at mid-summer in the periphery. Higher activity can increase the encounter rate and, therefore, the probability of detection. Finally, big patches can be expected to hold large populations, which might increase the probability of detecting a lizard. Thus, to find out the model that better explained detection probability, we built models with constant *p* and with all possible variable combinations among vegetation structure, day of survey and patch area as predictors of detection. Then, we compared models based on AIC and selected those with ΔAICc < 2 [49]. The model including the three variables was the best in the core, and the second best model in the periphery (ΔAIC = 0.38) (S4 Appendix). Consequently, all three variables were used as predictors of detection probability in all subsequent analysis in both regions.

In order to find out which were the relevant scales at which occupancy is explained in each region we tested whether occupancy patterns are explained at single scale(s) or simultaneously at multiple scales. Single-scale models included all composition variables measured at the same scale, plus configuration and patch variables, and multi-scale models included each composition variable at its scale of effect, together with configuration and patch variables. Therefore, before building multi-scale models we needed to find out which was the scale of effect of each composition variable -percentage of habitat, crops_pastures and urban- in each region. For this purpose, we fitted univariate models with each of these variables at each scale as predictor of occupancy ($p$) and selected the scale with the highest Nagelkerke $R^2$ ($RN^2$) as the scale of effect. In cases when the highest $RN^2$ value was present in several scales, the smallest scale was selected. For proximity index (prox), which is a scale-dependent configuration variable, the same procedure was applied to find its scale of effect in each region.

Then, to avoid collinearity among variables included in the same model, we applied a Spearman rank correlation test (S5 Appendix) to each single-scale and multi-scale dataset. Among correlated variables ($rs > 0.60$) we selected the one with the strongest effect on occupancy probability. Additionally, we calculated the variance inflation factor (vif) of selected covariates, and retained those with vif<10. In both regions we found strong collinearity among some variables that might have an important ecological role on occupancy. Therefore, in order to avoid skipping relevant variables from the analysis due to collinearity, we run several sets of single-scale and multi-scale models in each region (S6 Appendix). Each set included all non-correlated variables, and only one from the pair of correlated variables. In the core, Np_dist was correlated with prox at all scales, as well as crop_pas with urban. Both, Crop_pas and urban, might exert strong pressure on the occupancy, and proxy is a scale dependent measure of isolation that might have different explanatory power compared to Np_dist. Therefore, we run four sets of single-scale models for this region: Np_dist and crop_pas, Np_dist and urban, prox and crop_pas, or prox and urban. For the multi-scale model in the core, crop_pas was not correlated with urban; thus, both variables could be simultaneously included and only two multi-scale models were fitted, one with np_dist and one with prox. In the periphery, habitat was negatively correlated with urban at all scales, as well as in the multi-scale dataset. Therefore, for this region we fitted two single-scale models at each scale and two multi-scale models, one with habitat and the other with urban.

After having found the best model for detection probability, the scale of effect of composition based variables and prox to be used in multi-scale models, and having tested for collinearity among variables, we could then proceed with building single-scale and multi-scale global models. All global models were tested for Spatial Autocorrelation of Residuals ('SAC') to avoid underlying spatial processes to affect our results. For this, we calculated Global Moran's I and when significant SAC was found, an autocovariate parameter was calculated by means of principal components of neighbor matrices (PCNM) and added to the global model [50]. Goodness-of-fit test and overdispersion parameter (c-hat) were estimated by applying the parametric bootstrap procedure proposed by MacKenzie and Bailey [51] and implemented in the 'AICcModavg' package of R [52].

Finally, to find out the best model(s) explaining occupancy patterns in each region, we generated all possible models starting from each single-scale and multi-scale global model, with the function *dredge* of MuMiN package in R [53]. Then, we selected the models with ΔAIC < 2 [49]. Selected models were evaluated based on indicators that can be derived from a confusion matrix, which contains observed and predicted presence/absence (1/0) values of a given model [54]. We calculated the percent correctly classified (PCC), the area under the receiver operator characteristic curve (AUC) and Kappa statistics. All indicators have values ranging from 0 to 1. Kappa measures the agreement between the observed presence/absence values and those

expected by chance, and can be calculated at different thresholds used to translate predicted probabilities into 0/1 values. We calculated two Kappa measures, one at threshold of 0.5 (Kappa0.5) and another one at the optimized threshold (Kappaopt), where the optimized threshold was determined by calculating Kappa at each threshold from 0 to 1 at intervals of 0.01. All indicators were calculated with the 'PresenceAbsence' package of R [55]. Additionally, we also calculated the $RN^2$ of each selected model. We then selected the models with the highest value for most of the model indicators, and compared among all the single-scale models, and with the multi-scale models. Lastly, we determined which variables influenced the most occupancy patterns in each region, and whether the multi-scale models outperformed the single-scale models.

## Results

A total of 172 lizards were detected in both regions, 135 in the core and 37 in the Periphery. From 42 patches visited in the core, lizards were detected in 17 patches in both surveys and in 7 patches in one survey, for a total of 24 patches occupied. In the periphery, 7 out of 33 patches were occupied, and lizards were detected in 5 patches in both surveys and in 2 patches in one survey.

### Scales at which occupancy is explained in each region

The effect of composition-based variables (urban, crop_pas, habitat) and the proximity index (prox) on occupancy probability as single variables is shown in Fig 2. At all scales, the effect of urban, crop_pas and prox was higher in the periphery (Fig 2A) compared to the core (Fig 2B). At the core, crop_pas and prox showed a low, almost constant effect across scales, and the effect of urban at its scale of effect (50m) was just slightly higher compared to the other scales. By contrast, in the periphery the difference among scales was much more marked for these variables. Here, the scale of effect of urban was found at 500m, and the effects of crop_pas and prox at 1000m and 2000m, respectively, but their effects did not change considerably across scales. The effect of habitat at small scales (<500m) was similar between regions, but increased with scale in the periphery, reaching its maximum at 2000m, and decreased with scale in the core. The effect of natural covers that do not represent habitat in the periphery was strongest at large scales (Fig 2C). The effect of densewood showed a tendency to increase with scale up to 2000m, after which a slight decrease in the effect is found. A tendency to increase with scale was observed for humid_grass after 250m, reaching its peak at the scale of 3000m.

Although the scale of effect of individual composition-based variables was larger in the periphery compared to the core, when combining effects of multiple variables, representing not only landscape composition but also landscape configuration and patch characteristics, we found that the response of the species to habitat loss occurs at a much smaller spatial scale in the periphery relative to the core; (Table 2, Table 3). Specifically, the top single scale models explaining occupancy probability in the core were in the range of 500 m and higher (see S7 Appendix for best models selected at small scales), while in the periphery the best SS models were found already at 150 m.

### Most important variables at single scales

We found differences between regions regarding the variables that consistently had an effect on the occupancy probability across scales in SS models. In the core, most important variables were those defining landscape configuration and patch geometric characteristics (Table 2). Dist_river appeared consistently in all SS models, as well as a measure of isolation, either np_dist or prox. Perimeter and shape_index were also included in most models across scales.

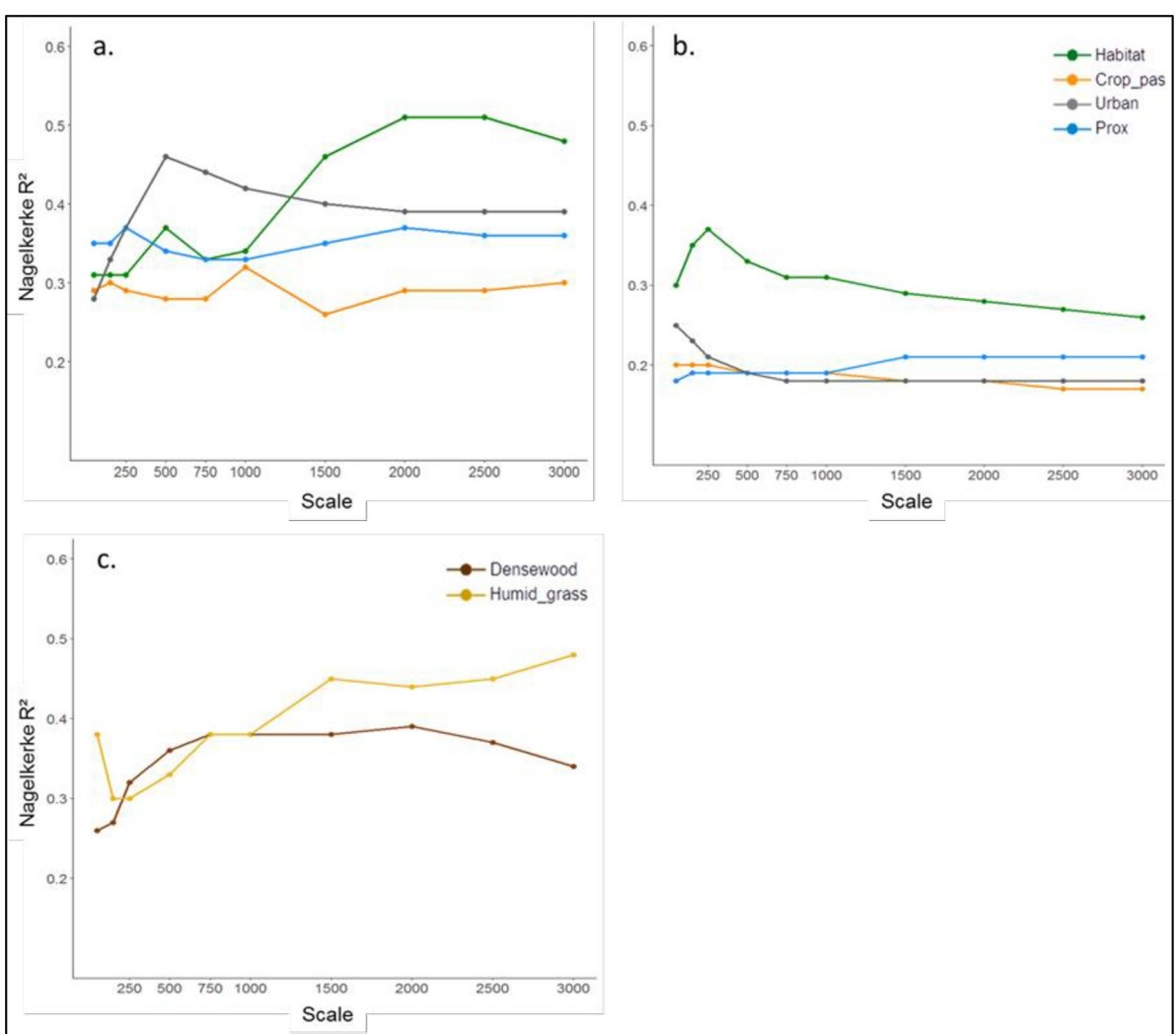

**Fig 2.** Effect of composition based variables and proximity index through spatial scales in the periphery (a, c) and in the core (b).

Area was not as commonly included as the variables mentioned above but was present in half of the SS models (15 out of 31), principally in models from 500m to 2000m. In the core, occupancy probability across single scales increased with isolation and perimeter and decreased with distance to the river, patch area and shape index. Although all indices across single scales had very close values, the best model was found at 750 m, which additionally included habitat, a variable that appeared only in few SS models. Prediction curves of the best model in the core showed that occupancy probability starts to decrease with a distance of 150m from the river, and reaches a value of 1 already with 10m distance from nearest patch and 20% of habitat coverage (Fig 3). Comparably, in the periphery, a combination of variables related to landscape composition, patch geometry and habitat quality defined the occupancy probability across single scales (Table 3). Densewood and crop_pas had a positive effect on occupancy and were present in the majority of SS models, as well as perimeter and slope. As in the core, area appeared in half of the SS models (13 out of 25), and was concentrated in scales above 500m, having a negative effect on occupancy probability. In the periphery, almost all indices had the same value across SS models. Based on the prediction curves, occupancy was above 0.5 when

**Table 2. Multiscale (ms) and single scale selected models at the core region.**

| Scale | $RN^2$ | PCC | AUC | Kappa0,5 | Kappaopt | Dist_river | Np_dist | Prox | Habitat | Crop_pas | Urban | Area | Perimeter | Shape_index | Veg_str | Radiation |
|---|---|---|---|---|---|---|---|---|---|---|---|---|---|---|---|---|
| ms | 0.4 | 0.761 | 0.824 | 0.513 | 0.559 | | | | + | | | | | | | |
| | 0.43 | 0.761 | 0.821 | 0.513 | 0.559 | | | | + | | | | | | | |
| | 0.33 | 0.761 | 0.821 | 0.513 | 0.513 | | | | + | | | | | | | |
| | 0.37 | 0.761 | 0.824 | 0.513 | 0.513 | | | | + | | | | | | | |
| 500 | 0.7 | 0.928 | 0.902 | 0.851 | 0.851 | - | | - | | | | - | + | - | + | - |
| | 0.7 | 0.928 | 0.902 | 0.851 | 0.851 | - | | - | | + | | - | + | - | + | |
| 750 | 0.68 | **0.952** | **0.918** | **0.901** | **0.901** | - | + | | + | | | | + | | | |
| | 0.62 | 0.928 | 0.878 | 0.851 | 0.851 | - | + | | + | | | + | + | | | |
| 1000 | 0.69 | 0.928 | 0.871 | 0.851 | 0.851 | - | + | | | | - | - | + | - | | |
| | 0.69 | 0.928 | 0.8855 | 0.851 | 0.851 | - | + | | | | - | - | + | - | | |
| | 0.7 | 0.928 | 0.902 | 0.851 | | - | + | | | | | - | + | - | | - |
| 1500 | 0.69 | 0.92 | 0.895 | 0.851 | 0.851 | - | + | | | | | + | + | - | | |
| | 0.68 | 0.928 | 0.868 | 0.851 | 0.851 | - | + | | | | - | - | + | - | | |
| | 0.7 | 0.928 | 0.902 | 0.851 | 0.851 | - | + | | | | | - | + | - | | - |
| | 0.66 | 0.928 | 0.895 | 0.851 | 0.851 | - | + | | + | | | | + | - | | |
| 2000 | 0.69 | 0.92 | 0.895 | 0.851 | 0.851 | - | + | | | | | - | + | - | | |
| | 0.67 | 0.928 | 0.895 | 0.851 | 0.851 | - | + | | | | - | - | + | - | | |
| | 0.7 | 0.928 | 0.902 | 0.851 | | - | + | | | | | - | + | - | | - |
| | 0.7 | 0.928 | 0.902 | 0.851 | 0.851 | - | + | | + | | | | + | - | | - |
| | 0.67 | 0.928 | 0.895 | 0.851 | 0.851 | - | + | | + | | | | + | - | | |
| | 0.67 | 0.928 | 0.902 | 0.851 | 0.851 | - | | - | + | | | | + | | + | - |
| 2500 | 0.69 | 0.928 | 0.891 | 0.851 | 0.851 | - | + | | + | | | | | | | |
| | 0.69 | 0.928 | 0.898 | 0.851 | 0.851 | - | | | + | | | | | | | - |
| | 0.67 | 0.928 | 0.895 | 0.851 | 0.851 | - | | - | | | - | | + | - | | |
| | 0.7 | 0.928 | 0.895 | 0.851 | 0.851 | - | | - | | | - | - | + | - | + | |
| | 0.69 | 0.928 | 0.895 | 0.851 | 0.851 | - | | - | | + | | | + | - | | |
| | 0.64 | 0.928 | 0.895 | 0.851 | 0.851 | - | | - | | + | | | + | - | | |
| | 0.63 | 0.928 | 0.895 | 0.851 | 0.851 | - | | - | | + | | | + | - | | |
| | 0.66 | 0.928 | 0.855 | 0.851 | 0.851 | - | | | - | + | | - | + | - | | |
| 3000 | 0.7 | 0.92 | 0.899 | 0.851 | 0.851 | - | + | | | | - | - | + | - | | |
| | 0.69 | 0.928 | 0.902 | 0.851 | 0.851 | - | + | | | | - | | + | - | | |
| | 0.69 | 0.928 | 0.895 | 0.851 | 0.851 | - | + | | | | | - | + | - | | |
| | 0.69 | 0.928 | 0.895 | 0.851 | 0.851 | - | | - | | | - | | + | - | | |
| | 0.68 | 0.928 | 0.895 | 0.851 | 0.851 | - | | - | | | - | | + | - | | |
| | 0.68 | 0.928 | 0.895 | 0.851 | 0.851 | - | | - | + | + | | | + | - | | |

Only variables explaining occupancy probability are presented, and the direction of their effects is shown as positive (+) or negative (-). Models with the same set of variables represent models with different combinations of the three variables explaining detection probability. In bold is signalized the model with the highest values for most of the model performance indicators.

the proportion of densewood was between 0.4 and 0.6 and the proportion of crop_pas between 0.3 and 0.7 (Fig 4).

Other variables had a lower representativeness across single scales in each region. In the core, the effects of composition-focused variables were mostly concentrated at larger scales. Urban was present in most of the models at 1000m and 3000m and crop_pas appeared in very few models, from which the majority belonged to the 2500m scale. Habitat also had a low representativeness in SS models in the core with most of them being at the 2000m and 2500m scales. Thus, habitat was not very consistent in explaining occupancy probability across scales

**Table 3. Multiscale (ms) and single scale selected models in the periphery region.**

| Scale | RN² | PCC | AUC | Kappa0,5 | Kappaopt | Dist_river | Np_dist | Prox | Habitat | Densewood | Humid_grass | Crop_pas | Urban | Area | Perimeter | Shape_index | Veg_str | Slope |
|---|---|---|---|---|---|---|---|---|---|---|---|---|---|---|---|---|---|---|
| Ms | 0.86 | 1 | 1 | 1 | 1 | | + | - | | + | | | | - | + | | | + |
| | 0.83 | 1 | 1 | 1 | 1 | + | | | | + | - | | - | | + | | - | |
| | 0.79 | 1 | 1 | 1 | 1 | | + | | | + | - | | | | + | | - | |
| 50 | 0.83 | 0.939 | 0.928 | 0.835 | 0.835 | | | | + | | - | + | | | + | | - | + |
| 150 | 0.85 | 1 | 1 | 1 | 1 | | | | | + | | + | | | + | | - | + |
| | 0.88 | 1 | 1 | 1 | | | | | | + | | + | | | + | | - | + |
| | 0.82 | 1 | 1 | 1 | | | | | - | + | | + | | | | - | | + |
| 250 | 0.86 | 1 | 1 | 1 | | - | + | | | + | | + | | - | + | | | + |
| | 0.85 | 1 | 1 | 1 | 1 | | | | | + | - | + | | | + | | - | + |
| | 0.85 | 1 | 1 | 1 | 1 | | | | - | + | | + | | | + | | - | + |
| | 0.84 | 1 | 1 | 1 | 1 | | | | | + | - | + | | | + | | | + |
| | 0.84 | 1 | 1 | 1 | 1 | | | | - | + | | + | | | + | - | | + |
| | 0.83 | 1 | 1 | 1 | 1 | | | | - | + | | + | | - | + | | | + |
| | 0.85 | 1 | 1 | 1 | 1 | | | | | + | | + | + | | + | | | + |
| | 0.84 | 1 | 1 | 1 | 1 | | | | | + | | + | + | - | + | - | | + |
| | 0.83 | 1 | 1 | 1 | 1 | | | | | | | + | + | - | + | | | + |
| 500 | 0.85 | 1 | 1 | 1 | | - | | - | | + | | + | + | - | + | | | + |
| 750 | 0.86 | 1 | 1 | 1 | 1 | - | | - | | + | | + | | - | + | | | + |
| | 0.79 | 1 | 1 | 1 | 1 | | | - | | | - | + | | - | + | | | + |
| | 0.83 | 1 | 1 | 1 | 1 | | | | | + | - | + | | - | + | | - | + |
| 1000 | 0.83 | 1 | 1 | 1 | 1 | | + | - | | + | | | | - | + | | - | + |
| | 0.8 | 1 | 1 | 1 | 1 | | + | - | | + | | | | - | + | | | + |
| 1500 | 0.85 | 1 | 1 | 1 | 1 | | + | - | | + | | | | - | + | | - | + |
| 2000 | 0.86 | 1 | 1 | 1 | 1 | | + | - | | + | | | | - | + | | | + |
| 2500 | 0.86 | 1 | 1 | 1 | 1 | | + | - | | + | | | | - | + | | | + |
| 3000 | 0.86 | 1 | 1 | 1 | 1 | | + | - | | + | | | | - | + | | | + |
| | 0.83 | 1 | 1 | 1 | 1 | | + | | | + | | | | - | + | | - | + |
| | 0.84 | 1 | 1 | 1 | 1 | + | | - | | + | - | | - | | + | | | + |

Only variables explaining occupancy probability are shown and the direction of variables' effect is marked as positive (+) or negative (-). Models with the same set of variables represent models with different combinations of the three variables explaining detection probability.

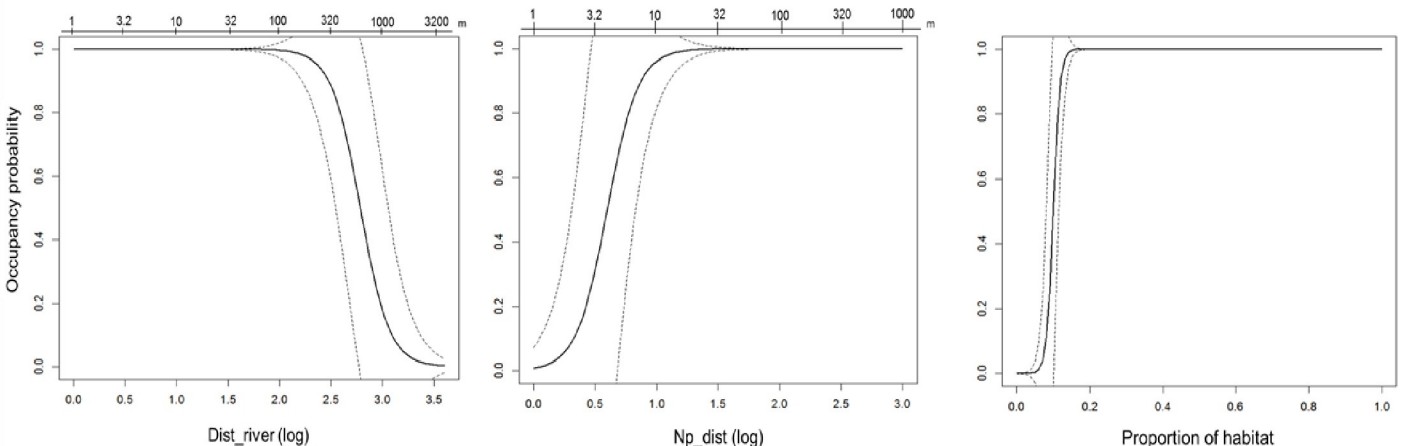

**Fig 3. Predicted occupancy probability as function of distance to river (Dist_river), distance to nearest patch (Np_dist), and proportion of habitat in the best model at scale 750m.** For Dist_river and Np_dist the x axis at the top represents distance values in meters.

in this region, despite being present in the best model at 750m. In the periphery, variables that appeared in much fewer models were np_dist, prox and veg_str. Isolation effects, either as np_dist or prox, were concentrated at large scales and appeared in all models above >1000m having a positive effect on occupancy. Veg_str was common in models at small scales (50-250m) and its effect on occupancy was negative.

## Multi-scale versus single-scale models

In the core region, when including composition-focused variables at its individual scale of effect in MS models, those with only habitat as predictor of occupancy probability performed better than models with any other combination of variables. However, in this region the best MS models did not outperform the best SS models at all scales for any of the model indicators (Table 2).

At the periphery, the performance of the MS models was equal to that of all SS models (Table 3). MS models in the periphery were partially similar to those in SS models, with

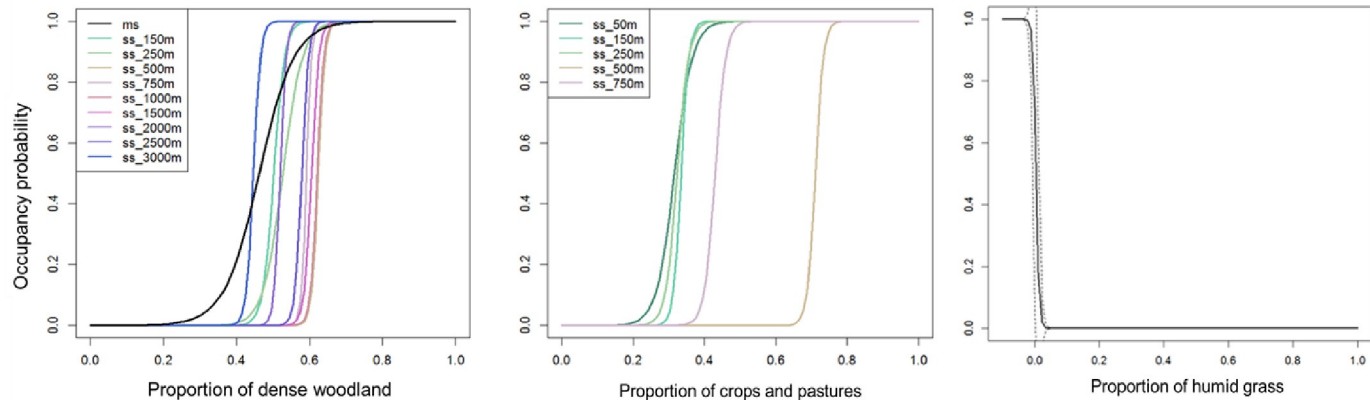

**Fig 4. Predicted occupancy probabilities in the periphery as function of the proportion of dense woodland and crops and pastures across single scales.** Probability curves plotted for each single scale (ss) correspond to the best model among the models in which the variable appears. Humid_grass curve correspond to the best MS model in which this variable was present.

densewood and perimeter still being very important and present in all MS models. Additionally, veg_str, np_dist and humid_grass were found to gain importance and were present in most of the MS models in the periphery. Humid_grass had a strong effect on occupancy probability, which dropped to zero at a very low coverage of this land cover class (Fig 4).

## Discussion

This study supports the hypothesis that the landscape structure and patch characteristics resulting from habitat loss affect differently the occupancy probability of *Lacerta viridis* in core versus peripheral populations. When comparing study areas with nearly similar landscape structure, we found that landscape composition had an overall stronger effect in the periphery compared to the core when land-cover classes were analyzed individually. In spite of the fact that the scale of effect of urban areas and crops and pastures was smaller in the core compared to the periphery, the effect of these variables was higher in the periphery at all scales (Fig 2). Similarly, the amount of habitat around patches had a stronger individual effect across all scales above 500m in the periphery compared to the core, while at smaller scales the strength of the effect was similar between regions. Therefore, our evaluation of the individual effects of landscape composition variables confirms the hypothesis of peripheral populations being more sensitive to habitat loss, not due to stronger effects appearing at smaller scales compared to the core, but because effects are stronger across all single scales.

Lower effects of individual landscape composition variables in the core compared to the periphery reflect what we found later in multivariate models: occupancy probability in the core was influenced by landscape configuration across single scales, whereas in the periphery occupancy was much more determined by landscape composition (Tables 2 and 3). The characteristics of the landscape surrounding a patch (patch context) affect occupancy probabilities mainly through their influence on the dispersal of individuals among patches [56, 57], an essential component for population persistence in structured landscapes [58]. Therefore, based on our study, it can be inferred that the most relevant parameter that might affect dispersal in the core is the spatial relation between patches and the river, while in the periphery our findings indicate a key parameter to be the proportion of different land covers in the matrix across scales. Thus, in the core, dispersal would be facilitated through connectivity defined by spatial configuration, while in the periphery it is overall landscape permeability that affects occupancy.

This sensitivity to matrix permeability in the periphery might be principally associated to the positive effect of crops and pastures on occupancy probability up to the 750 m scale (Table 3, Fig 4), suggesting that up to medium dispersal distances lizards in the periphery can cope with these land covers. Interestingly, although the variable crops and pastures was not ranked as important parameter in the core, when present in any model, its effect was positive as well (Table 2). The positive effect of crops and pastures on occupancy might be related with their effect on ecological processes that can occur during dispersal, like feeding, thermoregulation and predators avoidance [59, 60]. Despite higher exposure to predators, crops and pastures might offer food resources, as well as thermoregulation possibilities in the peripheral region, given a need for microhabitats with lower vegetation structure in this region. Open land covers might also be suitable for juvenile dispersal, as they are less conspicuous for predators than adults; and seasonal changes of crops might allow lizards to use different vegetation structures throughout the year. Moreover, age of individuals and vegetation structure of crops can have a positive interactive effect on the movement of some species. For example, in the case of the Franklin's ground squirrel (*Poliocitellus franklinii*), crops have been shown to have low resistance to movement, especially for juveniles in late summer and autumn, when

vegetation is higher and can hide them from predators [61, 62]. This might be especially important for the persistence of populations of lacertid species, in which juvenile dispersal is one of the most important dispersal events in life [63, 64], occurring precisely in late summer and autumn.

Tolerance to agricultural land cover might also be related with the maintenance of specific structures in the landscape that can increase the connectivity among populations, like vegetation in riparian zones, which are often inhabited by Lacertid species. For instance, the distribution of *Lacerta schreiberi* in Portugal was found not to be negatively affected by agriculture as long as vegetation along watercourses is maintained [65]. Our results suggest a similar finding in the core, with crops and pastures not having a negative effect and distance to river being one of the most important factors explaining population persistence (Table 2, Fig 3). Indeed, the vegetation at the banks of the Maritza River, as well as those of tributary rivers like the Tshaja river, is continuous along most of the river, thus potentially serving as an important corridor among patches. Hedges between fields are another landscape feature that might reduce the resistance of crops and pastures to the movement of lizards. Hedges were already found to play an important role for lizards at the community level, with cultivation patterns that include hedgerows sustaining higher species richness in a natural reserve in Cyprus [66]. Hence, in this region, the restoration of hedges around fields may improve connectivity and, with it, potentially occupancy probabilities. Regarding dense woodland in the periphery, which was present in all models in the periphery, its consistent positive effect (Table 3, Fig 4) might be due to the high correlation with prevalence of open woodland at all scales, which is one of the habitat types that *L. viridis* occupies in the periphery, rather than with permeability to dispersal.

In both regions, isolation had a positive effect on occupancy probability of habitat patches when combined with other variables. It was present across most single scales in the core but only above 1000 m in the periphery (Tables 2 and 3). Although the Island Biogeography Theory (IBT) [67] and meta-populations dynamics models [68] predict a negative effect of isolation, other conceptual models propose that the sensitivity to habitat configuration -isolation and patch area- vary depending on the overall amount of habitat in the landscape. The 'fragmentation threshold' hypothesis [69], for instance, states that habitat configuration is important when habitat amount is below ~30%; and the habitat amount hypothesis (HAH,[70]) postulates that due to a sample area effect, habitat configuration can perfectly be replaced by habitat amount surrounding the sampled site, with isolation having either any or positive effect (eg. [71]). However, in our study, habitat amount did not have a paramount effect on occupancy probability across single scales in any region, and therefore, the HAH does not apply to our case.

A conceptual model that could explain our results, is the one proposed by Villard and Metzger [72]. They propose that habitat configuration is important for the persistence of populations at intermediate levels of habitat amount. At low levels of habitat loss the species' density is high irrespective of isolation; as habitat loss proceeds, populations become dependent on configuration and dispersal among patches; finally at high levels of habitat loss -and subsequent increase in isolation- the species pool in the landscape has considerably decreased and populations' rescue is not possible anymore, even if connectivity is improved. The breadth and position of the range of intermediate values of habitat amount at which habitat configuration is important depends on the species sensitivity to both, habitat loss and habitat configuration. In our study, habitat amount was not important across models and isolation did not have a negative effect. Hence, following the model of Villard and Metzger ([72], fig 6-vi.), populations of *L. viridis* in the core and the periphery seem to have low sensitivity to both, habitat loss and isolation, which predicts a broad intermediate level that starts after considerable habitat loss.

In other words, lizards' populations seem to be able to cope with habitat loss independently of habitat configuration up to high levels of habitat reduction. With further habitat loss populations depend on configuration, but due to low sensitivity to it, they persist until considerable levels of isolation resulting from further reduction of habitat amount.

In terms of patch characteristics, the most important variable was perimeter, which was present in all models in both regions and had a positive effect on occupancy probability (Tables 2 and 3). Also, patch area was found in half of the models in each region, having in all but two cases a negative effect on occupancy probability. Positive perimeter effects coupled with negative effects of area are closely related to positive edge effects, [73], due to perimeter-to-area ratio increasing with decreasing area. In the core, additionally, shape index, which in our study increases with patch irregularity, had a negative effect on occupancy probability. Negative effects of shape index are related to decreasing core patch area [74, 75]. Thus, our results suggest that in the core occupancy probability might be influenced by positive edge effects together with sensitivity to core area, a pattern that has been found in species that use both, interior and patch edges [76]. Comparably, in the periphery, where shape index did not affected occupancy across scales, lizards might have preference for edges.

Differential preference in the use of patch edges between peripheral and core populations of *L. viridis* might result from differences in microhabitat selection between regions. In the periphery, overall radiation is lower compared to the core, and lizards compensate by selecting open microhabitats with low vegetation structure in order to maximize the exposure to radiation. In the core, where radiation and temperatures are higher, lizards select for microhabitats with higher vegetation structure that provide shadow and allow lizards to cool after basking hours. Thus, lizards in the periphery might use edge more often along the day and throughout the year, while in the core the preference of lizards for edges may correspond to basking hours in the early morning and late afternoon, and more often in early spring compared to late spring and summer. The relation between the effect of patch characteristics on occupancy probability and microhabitat selection and thermoregulatory behavior of lizards was also indicated by vegetation structure, which had a positive effect on occupancy probability in the core but a negative effect in the periphery. These results suggest that ecological processes at the individual level, like microhabitat selection and thermoregulation, might affect population persistence in the patch and generate occupancy patterns at the landscape scale.

Although vegetation structure was important for the occupancy probability in both regions in models at small scales (<500 m) (Tables 2 and 3), it was only in the periphery where another variable defining habitat quality, which is slope, was important across all single scales and retained in multi-scale models, suggesting a stronger dependency of peripheral populations of *L. viridis* on habitat quality when interacting with other parameters at multiple scales. Northern peripheral populations of *L. viridis* have a smaller niche size compared to core ones, which makes them more stenoecious or habitat specialist than core populations [24], a pattern also found in insects [26, 77], fishes [78] and other lizards [77]. Furthermore, habitat specialization is closely related with higher dependency on habitat quality [13], and occupancy probabilities have been found to be strongly influenced by habitat quality in specialist species of insects [79, 80], small mammals [81, 82] and lizards [83] inhabiting modified landscapes, in comparison with generalist species. In this regard, our study supports the existence of this pattern, but this time at the intraspecific level, with populations differing in their degree of habitat specialization depending on their geographic position in the distribution range of the species.

Several studies have linked the position in the distribution range with vulnerability to extinction, and point out that peripheral populations might be at higher risk of local extinction [84, 85]. Moreover, specific traits of peripheral populations, like lower abundance [86], lower genetic variability [87, 88] and smaller niche [78, 89], have been proposed to explain its higher

vulnerability. Position in the range and vulnerability of extinction have also been linked with sensitivity to human modified landscapes (e.g. [90]), and extensive multispecies approaches have demonstrated higher sensitivity to habitat loss of peripheral populations in the Palearctic region [91]. However, only very few studies have made the complete link between position in the range, species traits and vulnerability of extinction in modified landscapes. For instance, [92] found that peripheral populations of the lizard *Lacerta agilis* had a lower genetic variability and also a higher sensitivity to patch size, compared to core populations. In this context, our work also throws some light upon the possible ecological mechanisms behind the relationship between position in the range, sensitivity to habitat loss and populations' traits, by identifying the parameters of landscape structure and patch characteristics to which northern peripheral and more specialized populations of a broad ranging species are more sensitive compared to core populations.

With respect to the analysis performed and the model selection procedures, it is important to note that the high values obtained for model evaluation indices in all of our models, can be strongly related to the fact that we tried as much as possible to cover the range and type of variables that might influence occupancy. Also, it might be strongly related to the model selection procedure that we applied, in which models were first selected based on $\Delta AIC < 2$ and then, from this group of best models, we selected those with the highest values for the indices evaluated. High indices values indicate that the models can discriminate very well between patches where the lizard is present and those where it isn't, which in a binary classification scheme can be expected for models that explain also high levels of variance ($> 63\%$ in the single scale models in the core, and $>79\%$ in all models in the periphery), and thus, our results highlight even more the fact that the inclusion of specific variables (the most common ones found in the models) might be important for model accuracy. In the periphery very high indices values of selected models ($= 1$) might also be due to the fact that the majority of the patches in the sample were not occupied, and then, the classification ability is higher. However, given models presented in the results represent an extremely reduced group among all the model initially run, we still consider that the predictive and classification abilities of selected models, by themselves and not due to sample distribution, is very high.

An additional important remark regarding models' output, is that in multivariate models the direction of the effect of each variable can change depending on other variables present in the model [93]. For some of the variables that we considered, like isolation in the core and crops and pasture in both regions, the individual effect was negative (S8 Appendix), but in combination with other variables the effect was positive. Positive effects of these variables were systematic in all multivariate selected models where these variables were present, and therefore, we rely on our results, and highlight the importance of testing coefficients direction when variables are alone or in combination with other variables.

Regarding the land cover classification approach that we apply, it is important to consider that although the ideal methodological approach to compare among landscapes is to produce classified maps with data obtained from the same source, our approach was perfectly sufficient to perform the ecological analysis that we carried out. As stated by Fynn and Campbell [94], possible shortcomings of landscape ecology studies using imagery from different sources might come out in cases when images with coarse resolution are compared with finer resolution imagery. However, in our study the resolution of both, the IPR map used for the periphery and the rapid eye satellite imagery used for the core, was the same (5m), and additional information used for the classification in both regions had the same source (Urban atlas, TCD and imperviousness layers of CORINE) and resolution; orthophotos used for some parts of the map in the periphery were rectified by IPR and had also a very high accuracy. Dissimilar sources of information might as well represent a disadvantage due to the different methods

used for the classification process in the IPR map of the periphery compared to those we applied to the Rapid Eye satellite imagery in the core region. However, given the high specificity of the original classification of the IPR map (> 60 classes), which we afterwards reclassified in broader classes, we consider that the output of both maps had similar accuracy (>90%), and therefore, perfectly allowed to compare between landscapes and precisely calculate percentages of land cover classes. Comparability was also achieved through careful examination of maps by the first coauthor who knows both study sites extremely well after having spent several months in both regions, and therefore had trustable on-the-ground information, and by means of thorough and systematic application of specific criteria to classify each land cover in both regions (Table 1).

### Implications for conservation measures

In the periphery, the most important was the landscape composition and the permeability represent mostly by the presence of crops and pastures. Our results show that this effects are present already at very low scales, and that in scales between 50 to 500m occupancy probability increases already over 0.8 with percentages of crops and pastures between 30 to 40%. On the other side this permeability decreases very fast with already a low percentage of humid grasses. Therefore, we strongly recommend to increase matrix permeability by applying a more heterogeneous cultivation pattern that includes hedges and line structures with vegetation corresponding to the habitat of the species, as well as the inclusion of such structures through areas with humid grassland.

With respect to patch characteristics, it is very important to increase the availability of edge in the patches. This can be achieved by increasing patch size with linear structures to maintain a high perimeter to area ratio. In parallel, these linear structures can also serve to connect through the agricultural landscape. Finally, maintaining high levels of habitat quality is also very important in this region, and can be achieved by keeping low levels of vegetation structure, and specially by protecting valley's slopes from overgrown vegetation. Similarly, overgrown vegetation should be avoided in open woodlands, which are usually located in slopes and at the borders of dense woodland areas.

In the core, regarding configuration of the landscape, the most important management measures are, first to protect the patches that are close to the river, or at the riverside, and second to structurally connect with the river those patches that are further. Based on our analysis, patches with a distance to the river lower than 320m have an occupancy probabilities over 0.8, and patches with distances longer than ~650 m have probabilities lower than 0.5. Then, we recommend to protect -and restore where necessary- the river bank vegetation along the Maritza River and its tributary rivers, as this areas might act as important corridors for the species, and to connect further habitat remnants with this large riverside corridor, through additional structures with habitat vegetation. As in the periphery, hedges and habitat lines surrounding crops could improve connectivity through the landscape.

Regarding composition, it is very important to protect the habitat surrounding patches, principally at a scale of 250m, which is the scale of effect of this land cover and at which the variable was included in multi-scale models, being the only variable present in these models. Habitat was also present in the best model at the scale of 750m, and our results show that with only a small increment in the percentage of habitat at this scale (~ 10%) the probability of occupancy substantially increases (Fig 3).

With respect to patch characteristics, we found that the shape of the patches is very important for both, maintaining a large perimeter and also sufficient core area. Therefore, we strongly recommend to not alter the shape of remnant patches that already have a regular

shape, and to restore habitat at the direct borders of patches with irregular shapes, in order to increase perimeter and core area by 'softing' angular and irregular shapes. Regarding habitat quality, we suggest to protect the vegetation structure in remnant patches, avoiding practices that can diminish it. This means, maintaining different vegetation levels that include grasses, shrubs, rocks, fallen trunks, trees, etc. Grazing, for instance, can have a very rapid negative effect in the quality of the patches by substantially reducing vegetation structure (pers. observation), given cows and goats feed on the low and medium strata, and goats also lower branches of woody plants. As a consequence lizards lose refuge and structures to bask. Also, as vegetation structure decreases radiation incidence increases, consequently augmenting temperatures and diminishing humidity, with the habitat becoming drier and less suitable for the species.

In both regions we recommend to monitor the populations. Further insights in the abundance and condition status of individuals would be very useful to more deeply asses the status of populations.

## Conclusions

Our study shows that northern peripheral, more specialized populations of *L. viridis* are also more sensible to the effects that habitat loss has on the landscape structure and on the characteristics of remnant habitat patches. In comparison with populations in the core, the occupancy probability of populations in the periphery was found to be more affected by landscape composition, which suggests substantial dependency on matrix permeability; also, habitat quality had a stronger influence on populations in the periphery and our results regarding patch geometric characteristics in this region suggest a preference of the species for patches with more edge in relation to patch core area. Comparably, in the core, we found that persistence of populations is mainly affected by the possible connectivity that the river bank vegetation offers through the landscape. Also, the species in this region seems to be an omnipresent species regarding its use of the patch, requiring both long edges and also enough core area in the interior of the patch. Finally, in both regions the species had low sensitivity to habitat amount and to habitat configuration, an outcome that strongly differs from the expectations of the IBT, the meta-populations dynamic models and also from the HAH, but one that fits conceptual–and empirically tested–models that describe a more gradual relationship between habitat amount and isolation.

## Supporting information

**S1 Appendix. Distribution of variables representative of habitat configuration in each region.**
(DOCX)

**S2 Appendix. Location of visited patches in each region.**
(DOCX)

**S3 Appendix. Maps of classified land cover classes in each region.**
(DOCX)

**S4 Appendix. Models for detection probability in each region.** Variables were included to model only detection probability (p) while maintaining occupancy probability (psi) constant.
(DOCX)

**S5 Appendix. Spearman rank correlations of no scale dependent variables in each region.**
(DOCX)

**S6 Appendix. Different sets of models ran in each single scale and multiscale models in each region.** Each set includes all variables plus the variable or combination of variables indicated.
(DOCX)

**S7 Appendix. Best selected models at small scales from 50m to 250m in the core region.**
(DOCX)

**S8 Appendix. Individual effects of non-scale and scale dependent variables.**
(DOCX)

## Acknowledgments

We thank Pavel Stoev and Nikolay Tzankov (RIP) for supporting the field work in Plovdiv (Bulgaria) and for providing information about the species in this region. We thank Jiri Moravec for facilitating information about *L. viridis* in Prague, and to Jan Chmelar for his help identifying populations around Prague.

## Author Contributions

**Conceptualization:** Ana María Prieto-Ramirez, Guy Pe'er, Dennis Rödder, Klaus Henle.

**Data curation:** Ana María Prieto-Ramirez.

**Formal analysis:** Ana María Prieto-Ramirez.

**Funding acquisition:** Ana María Prieto-Ramirez.

**Investigation:** Ana María Prieto-Ramirez.

**Methodology:** Ana María Prieto-Ramirez, Leonie Röhler, Anna F. Cord.

**Resources:** Klaus Henle.

**Software:** Leonie Röhler, Anna F. Cord.

**Supervision:** Guy Pe'er, Dennis Rödder, Klaus Henle.

**Visualization:** Leonie Röhler, Anna F. Cord.

**Writing – original draft:** Ana María Prieto-Ramirez.

**Writing – review & editing:** Guy Pe'er, Dennis Rödder, Klaus Henle.

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
