## [Decision Letter · Decision Letter 0]

13 Nov 2019

PONE-D-19-24402

Differential effects of habitat loss on occupancy patterns of the eastern green lizard Lacerta viridis at the core and periphery of its distribution range

PLOS ONE

Dear Prieto Ramírez,

Thank you for submitting your manuscript to PLOS ONE. After careful consideration, we feel that it has merit but does not fully meet PLOS ONE’s publication criteria as it currently stands. Therefore, we invite you to submit a revised version of the manuscript that addresses the points raised during the review process.

In Addition to the comments of the reviewer (see below), which I fully agree with, please correct the following:

- Table 1: words start either with capital or small letters, obviously without rule. Use one way of writing only.

- Line 247: herbs are classified <30 or >40cm, <80 or >90 cm. What About plants which are between 30 and 40 or between 80 and 90cm?

- Line 286: In order to find out which were the relevant scales

- Results: You must indicate the number of observations done.

- Line 458: our Evaluation (delete 'the')

- Lines 468-470. You speak about dispersal, but could you observe dispersal directly, or is a guess or inference?

- Lines 560-561: I don't see that an example from Madagascar applies here. Delete it or take another.

- Line 568: provide (delete 's')

We would appreciate receiving your revised manuscript by Dec 28 2019 11:59PM. To enhance the reproducibility of your results, we recommend that if applicable you deposit your laboratory protocols in protocols.io, where a protocol can be assigned its own identifier (DOI) such that it can be cited independently in the future. For instructions see: http://journals.plos.org/plosone/s/submission-guidelines#loc-laboratory-protocols

We look forward to receiving your revised manuscript.

Kind regards,

Ulrich Joger

Academic Editor

PLOS ONE

Journal Requirements:

2. We note that [Figure 1] in your submission contains map/satellite images which may be copyrighted. All PLOS content is published under the Creative Commons Attribution License (CC BY 4.0), which means that the manuscript, images, and Supporting Information files will be freely available online, and any third party is permitted to access, download, copy, distribute, and use these materials in any way, even commercially, with proper attribution. For these reasons, we cannot publish previously copyrighted maps or satellite images created using proprietary data, such as Google software (Google Maps, Street View, and Earth). For more information, see our copyright guidelines: http://journals.plos.org/plosone/s/licenses-and-copyright.

You may seek permission from the original copyright holder of Figure [1] to publish the content specifically under the CC BY 4.0 license. 

If you are unable to obtain permission from the original copyright holder to publish these figures under the CC BY 4.0 license or if the copyright holder’s requirements are incompatible with the CC BY 4.0 license, please either i) remove the figure or ii) supply a replacement figure that complies with the CC BY 4.0 license. Please check copyright information on all replacement figures and update the figure caption with source information. If applicable, please specify in the figure caption text when a figure is similar but not identical to the original image and is therefore for illustrative purposes only.

3. In your Methods section, please provide additional location information of the study areas, including geographic coordinates for the data set if available.

4. In your Methods section, please provide additional information regarding the permits you obtained for the work. Please ensure you have included the full name of the authority that approved the field site access and, if no permits were required, a brief statement explaining why.

Reviewers' comments:

Reviewer's Responses to Questions

**Comments to the Author**

1. Is the manuscript technically sound, and do the data support the conclusions?

Reviewer #1: Partly

2. Has the statistical analysis been performed appropriately and rigorously? 

Reviewer #1: Yes

3. Have the authors made all data underlying the findings in their manuscript fully available?

Reviewer #1: No

4. Is the manuscript presented in an intelligible fashion and written in standard English?

Reviewer #1: Yes

5. Review Comments to the Author

Reviewer #1: The paper PONE-D-19-24402 submitted by Prieto-Ramirez et al for consideration in PLOS ONE examines the impact of habitat loss on space occupancy patterns in Lacerta viridis and compares such effects between a core and a peripheral area of the distribution range of the species. This is an interesting study, which aims at tackling a very relevant subject, i.e. how species respond to habitat fragmentation and whether these responses differ depending on whether we are dealing with a central or border population. The authors perform very thorough (although at times difficult to follow!) analyses and provide evidence that support the idea that different ecological responses drive space occupancy in the two regions and at different spatial scales. This is a study that provides interesting insights to the spatial ecology of the species, with relevance for conservation.

The manuscript is very well designed and set up. The authors provide a very thorough revision of the concepts and ideas used to establish their hypotheses and focus of interest, with what seems to be quite a complete review of the literature, although some parts of the manuscript will definitely need better literature support (see below).

Generally, I believe that the analyses conducted are robust and the conclusions made match the results obtained. However, there are several points that I believe need clarification, as the statistical framework used is quite complex and the descriptions provided are very dense. In addition, I have a serious doubt about the way how land cover classified maps were obtained, which should at least be considered explicitly. I detail the main points I believe need amendments below:

1. Regarding land cover classifications, the description of the methods used (p. 9, l. 197 – p. 10. l. 220) seems to suggest that at least three different methods were used, i.e. differing between the core and periphery areas; and in addition differing for the northern and southern regions of the periphery area (i.e. lines 217-220). I believe that this might make the data retrieved in different ways non comparable. As this is a key point for the study, I think the authors should modify their approach if possible to make land cover classifications uniform across areas, or at least be very clear about this point and consider its possible effects.

2. The choice of retaining models with deltaAIC<2 needs to be supported with literature (p. 13, l. 280 – 281).

3. The methods section is VERY dense, and it becomes difficult to follow for non-specialists; on the other hand it is efficient in being very detailed and providing a very thorough description of the procedures followed. To facilitate the reader, I recommend that the authors review this section to provide more clear links between each statistical procedure and its aim, i.e. starting and/or ending paragraphs with a clear statement of the objective of each analytical step.

4. Table 3: I find it VERY curious, even possibly flawed, that all PCC, AUC, and almost all Kappa values are not only identical, but actually equal to unit here!!! I am not a specialist on the software used for data analyses, but in statistics round 1s are either easily justified or the outcome of an error. I strongly recommend that the authors carefully review and clarify the cause of this result.

5. Throughout the discussion, I recommend that the authors review their text to provide direct links between their statements and the results on which these are based (i.e. in the form of references to Figures and Tables).

6. As a final point, I would like to see a more thorough treatment in the section of “Implications for conservation measures” (p. 30). This paragraph seems very basic and obvious, given the complexity of the analyses conducted in this study.

7. Throughout, please provide adequate referencing of all software used (e.g. p.7, l. 158 and elsewhere for ArcMap; p. 12, l. 269 for R; etc).

6. PLOS authors have the option to publish the peer review history of their article (what does this mean?). If published, this will include your full peer review and any attached files.

Reviewer #1: No

---

## [Author Response · Author response to Decision Letter 0]

28 Dec 2019

Note of the authors: The page and line numbering indicated in the responses correspond to the ‘Revised Manuscript with Track changes” version.

Comments of the Academic Editor

In Addition to the comments of the reviewer (see below), which I fully agree with, please correct the following:

- Table 1: words start either with capital or small letters, obviously without rule. Use one way of writing only.

Response: Corrected as suggested. (P.9, l. 198)

- Line 247: herbs are classified <30 or >40cm, <80 or >90 cm. What About plants which are between 30 and 40 or between 80 and 90cm?

Response: The data on vegetation structure that we collected was composed by two elements, height and percentage covered. We classified the vegetation of a given portion of the plot covered with the same vegetation type without interruption (eg. grass) based on the height of the vast majority of the covered portion (eg. if only some spikes of grass had height > 30 cm, but most of the area covered had <30 cm, then it was classified as herbs < 30cm). It was never a problem to classify vegetation under these categories, and thanks to this distinctive classification, we can assure that classes represent actual height differences.

- Line 286: In order to find out which were the relevant scales

Response: Corrected as suggested (P. 13, l. 292)

- Results: You must indicate the number of observations done.

Response: At the beginning of the Results section we added a paragraph with the basic information about number of lizards detected and number of patches occupied in each region. (P.16, l.360)

- Line 458: our Evaluation (delete 'the')

Response: Deleted (P. 25, l. 485)

- Lines 468-470. You speak about dispersal, but could you observe dispersal directly, or is a guess or inference?

Response: It is an inference based on the statement that landscape structure affects occupancy mainly through its effect on dispersal, which is widely supported in the literature, and for which we cited the most relevant literature:

“The characteristics of the landscape surrounding a patch (patch context) affect occupancy probabilities mainly through their influence on the dispersal of individuals among patches (48, 49), an essential component for population persistence in structured landscapes (50).” (P. 25, l.494).

In the next sentences we modified the text to make explicit the fact that this is an inference, by adding the text in red:

“Therefore, based on our study, it can be inferred that the most relevant parameter that might affect dispersal in the core is the spatial relation between patches and the river, while in the periphery our findings indicate a key parameter to be the proportion of different land covers in the matrix across scales. Thus, in the core, dispersal would be facilitated through connectivity defined by spatial configuration, while in the periphery it is overall landscape permeability that affects occupancy.” (P. 25, l 497)

- Lines 560-561: I don't see that an example from Madagascar applies here. Delete it or take another.

Response: The example was deleted.

- Line 568: provide (delete 's')

Response: Deleted.

Journal Requirements:

 Please ensure that your manuscript meets PLOS ONE's style requirements, including those for file naming. The PLOS ONE style templates can be found at…

Response: All formatting requirements were checked prior to this submission.

2. We note that [Figure 1] in your submission contains map/satellite images …

Response: The first map (Figure 1a) corresponds to the distribution range map of the species provided by the IUCN in its webpage. The corresponding Content Permission Form is attached under “Other” documents.

The other two maps were changed for Sentinel-2 cloudless images from 2016 provided by EOX IT Services GmbH under CCBY 4.0 license (https://s2maps.eu/).

Both references, to IUCN and to EOX IT Services GmbH were added in the caption of Figure 1 (P. 7, l. 144).

3. In your Methods section, please provide additional location information of the study areas, including geographic coordinates for the data set if available.

Response: Specific location information of the boundaries of the polygons that enclose the study areas in each region was added in the caption of Figure 1 (P. 6, l. 140). Additionally, a table with the coordinates of each visited patch was included as supplementary material (S1 Appendix 2). A reference to this table was introduced in the Field survey subsection of the Methods section (P.7, l. 160).

4. In your Methods section, please provide additional information regarding the permits you obtained for the work. Please ensure you have included the full name of the authority that approved the field site access and, if no permits were required, a brief statement explaining why.

Response: We added the following statement at the end of the Field Survey subsection in the Methods section (P.8, l. 180):

“As surveys were based on visual identification of lizards, and no collection of biological material or handling of animals was required, no permits were necessary for carrying out this study”.

Response: Many thanks for the clarification. We will provide DOIs to access our data after acceptance of the manuscript. We don’t wish to make any changes to our Data Availability statement.

Review Comments to the Author

Reviewer #1: 

The paper PONE-D-19-24402 submitted by Prieto-Ramirez et al for consideration in PLOS ONE examines the impact of habitat loss on space occupancy patterns in Lacerta viridis and compares such effects between a core and a peripheral area of the distribution range of the species. This is an interesting study, which aims at tackling a very relevant subject, i.e. how species respond to habitat fragmentation and whether these responses differ depending on whether we are dealing with a central or border population. The authors perform very thorough (although at times difficult to follow!) analyses and provide evidence that support the idea that different ecological responses drive space occupancy in the two regions and at different spatial scales. This is a study that provides interesting insights to the spatial ecology of the species, with relevance for conservation.

The manuscript is very well designed and set up. The authors provide a very thorough revision of the concepts and ideas used to establish their hypotheses and focus of interest, with what seems to be quite a complete review of the literature, although some parts of the manuscript will definitely need better literature support (see below).

Generally, I believe that the analyses conducted are robust and the conclusions made match the results obtained. However, there are several points that I believe need clarification, as the statistical framework used is quite complex and the descriptions provided are very dense. In addition, I have a serious doubt about the way how land cover classified maps were obtained, which should at least be considered explicitly. I detail the main points I believe need amendments below:

Response: We thank the reviewer very much for her/his appreciations about our work, as well as for the constructive comments that she/he has done. We are sure that they have substantially improve the quality of our manuscript.

1. Regarding land cover classifications, the description of the methods used (p. 9, l. 197 – p. 10. l. 220) seems to suggest that at least three different methods were used, i.e. differing between the core and periphery areas; and in addition differing for the northern and southern regions of the periphery area (i.e. lines 217-220). I believe that this might make the data retrieved in different ways non comparable. As this is a key point for the study, I think the authors should modify their approach if possible to make land cover classifications uniform across areas, or at least be very clear about this point and consider its possible effects.

Response: We fully agree that using different methods to obtain the land cover information is not the ideal approach. However, developing our own classified products for all sites based on satellite imagery was beyond the scope of the study. We had initially planned to use readily available products for both sites (as done e.g. in Adhikari and Hansen 2018, https://www.sciencedirect.com/science/article/pii/S0169204618302603). The lack of spatially and thematically detailed enough land cover data for the study site in the core (Bulgaria), however, required us to develop our own classification approach. In order to make the datasets as comparable as possible, we therefore used harmonized land cover classes (see Table 1) across the different study regions and datasets. All land cover maps were carefully evaluated by the first author who had spent several weeks in the field, had collected land cover ground true data and knows both study sites extremely well.

Following your suggestion, we have now added a paragraph that clearly states the use of different land cover information sources in our study as possible shortcoming and how we solved it in the Discussion (P.33, l. 670).

2.The choice of retaining models with deltaAIC<2 needs to be supported with literature (p. 13, l. 280 – 281).

Response: Corresponding reference was cited (P.13, l. 287; P. 15, l. 341)

49. Burnham KP, Anderson DR. Model selection and multimodel inference: A practical information-theoretic approach 2nd edition ed. New York: Springer-Verlag; 2002.

3. The methods section is VERY dense, and it becomes difficult to follow for non-specialists; on the other hand it is efficient in being very detailed and providing a very thorough description of the procedures followed. To facilitate the reader, I recommend that the authors review this section to provide more clear links between each statistical procedure and its aim, i.e. starting and/or ending paragraphs with a clear statement of the objective of each analytical step.

Response: We thank the reviewer very much for this comment. As suggested, we modified the methods, especially the statistical analysis subsection (P.12, l. 270), by clearly stating the objective of each step. We consider that the description of the analysis is now clearer and easier to follow.

4. Table 3: I find it VERY curious, even possibly flawed, that all PCC, AUC, and almost all Kappa values are not only identical, but actually equal to unit here!!! I am not a specialist on the software used for data analyses, but in statistics round 1s are either easily justified or the outcome of an error. I strongly recommend that the authors carefully review and clarify the cause of this result.

Response: 

1. Evaluated models presented in the results (Table 2 and Table3) are the result of two selection steps: an initial step, in which all models with ΔAIC < 2 were selected, and theoretically these models are equally ‘good’; and then, among these models we selected those with the highest indices values. Then, it is not so surprising that models evaluated into each single-scale and multi-scale dataset, have similar or the same values for all indicators, and that these values are high.

2. Also, all models presented in the results don’t come from the same datasets, even if they belong to the same single scale, due to the approach of generating different datasets per scale in order to avoid skipping important variables that presented high collinearity (S1 Appendix 6). Compiling together the best models of each dataset (after the two steps selection process mentioned above), makes it probable that values are repeated.

3. Secondly, PCC, Kappa, and AUC assess model discriminatory ability, that is, how accurately can the model divide between patches depending on presences/absence of the species. R² by its side, reflects how much of the variance is explained by the model and it’s predictive ability. All models from single scales in the core, and single scales and multi scale in the periphery had very high R² values (>60% and > 79% respectively). For such models explaining a high percentage of variance, it can be expected also high discrimination ability for threshold dependent indices (Kappa0.5 and Kappaopt) and threshold independent measures (AUC), reflected also in a high percentage of correct classification (PCC).

4. The fact that the majority of the patches in the periphery were unoccupied might have an effect on the classification ability of models. However, given the models presented in the results are an extremely reduced group of models selected among all possible variables combinations at all scales, we don’t have doubts on the high predictive and classification abilities of the models.

We had already shortly discussed this issue but it surely was not specific enough. Therefore, following the suggestion of the reviewer, we did a more profound discussion on this topic, adding the above mentioned arguments (P. 31, l.644)

5. Throughout the discussion, I recommend that the authors review their text to provide direct links between their statements and the results on which these are based (i.e. in the form of references to Figures and Tables).

Response: We followed the suggestion and included references to tables and figures, when necessary.

6. As a final point, I would like to see a more thorough treatment in the section of “Implications for conservation measures” (p. 30). This paragraph seems very basic and obvious, given the complexity of the analyses conducted in this study.

Response: We followed the suggestion and increase the accuracy of the recommendations that we have already expressed, by including specific information regarding percentages of land cover classes, distances and effects on occupancy probabilities (P.34, l. 694).

7. Throughout, please provide adequate referencing of all software used (e.g. p.7, l. 158 and elsewhere for ArcMap; p. 12, l. 269 for R; etc).

Response: References for all software and packages used were added.

---

## [Decision Letter · Decision Letter 1]

11 Feb 2020

Differential effects of habitat loss on occupancy patterns of the eastern green lizard Lacerta viridis at the core and periphery of its distribution range

PONE-D-19-24402R1

Dear Dr. Prieto Ramírez,

We are pleased to inform you that your manuscript has been judged scientifically suitable for publication and will be formally accepted for publication once it complies with all outstanding technical requirements.

With kind regards,

Ulrich Joger

Academic Editor

PLOS ONE

Additional Editor Comments (optional):

Reviewers' comments:

Reviewer's Responses to Questions

**Comments to the Author**

1. If the authors have adequately addressed your comments raised in a previous round of review and you feel that this manuscript is now acceptable for publication, you may indicate that here to bypass the “Comments to the Author” section, enter your conflict of interest statement in the “Confidential to Editor” section, and submit your "Accept" recommendation.

Reviewer #1: All comments have been addressed

2. Is the manuscript technically sound, and do the data support the conclusions?

Reviewer #1: Yes

3. Has the statistical analysis been performed appropriately and rigorously? 

Reviewer #1: Yes

4. Have the authors made all data underlying the findings in their manuscript fully available?

Reviewer #1: Yes

5. Is the manuscript presented in an intelligible fashion and written in standard English?

Reviewer #1: Yes

6. Review Comments to the Author

Reviewer #1: I have no further comments, great job integrating the previous ones, I´m happy to see this manuscript published.

7. PLOS authors have the option to publish the peer review history of their article (what does this mean?). If published, this will include your full peer review and any attached files.

Reviewer #1: No

---

## [Editor Report · Acceptance letter]

24 Feb 2020

PONE-D-19-24402R1 

Differential effects of habitat loss on occupancy patterns of the eastern green lizard *Lacerta viridis* at the core and periphery of its distribution range 

Dear Dr. Prieto Ramírez:

I am pleased to inform you that your manuscript has been deemed suitable for publication in PLOS ONE. Congratulations! Your manuscript is now with our production department. 

With kind regards,

on behalf of

Dr. Ulrich Joger 

Academic Editor

PLOS ONE